# Newly Diagnosed Crohn’s Disease Patients in India and Israel Display Distinct Presentations and Serological Markers: Insights from Prospective Cohorts

**DOI:** 10.3390/jcm11236899

**Published:** 2022-11-22

**Authors:** Idan Goren, Tali Sharar Fischler, Henit Yanai, Partha Pal, Bhargavi Adigopula, Sushmitha Pendyala, Girish Ganesh, Ravikanth Vishnubhotla, Keren Masha Rabinowitz, Efrat Shaham Barda, Durga Yadamreddy, Lihi Godny, Noam Peleg, Rupa Banerjee, Iris Dotan

**Affiliations:** 1IBD Center, Division of Gastroenterology, Rabin Medical Center Affiliated to the Sackler Faculty of Medicine, Tel Aviv University, Petah Tikva 49100, Israel; 2Department of Inflammation and Immunity, Lerner Research Institute, Cleveland Clinic, Cleveland, OH 44195, USA; 3Department of Gastroenterology, Asian Institute of Gastroenterology, Hyderabad 501301, India; 4Department of Genomics and Molecular Biology, Institute of Translational Research, Asian Institute of Gastroenterology and AIG Hospitals, Hyderabad 500032, India; 5Felsenstein Medical Research Center, Rabin Medical Center, Beilinson Hospital Affiliated to the Sackler Faculty of Medicine, Tel Aviv University, Petah Tikva 49414, Israel

**Keywords:** Crohn’s disease, inception cohort, India, complication, serology, prediction, genetic, East–West

## Abstract

Background: Crohn’s disease (CD) incidence is rising in India. However, features of newly diagnosed patients with CD in this population are largely unknown. The Indo-Israeli IBD GastroEnterology paRtnership (TiiiGER) aimed to investigate differences in presentation among patients with newly diagnosed CD in India and Israel, and to explore phenotype–serotype correlations. Methods: A prospective observational cohort study of consecutive adults (>18 years) conducted in two large referral centers in India and Israel (2014–2018). Clinical data, an antiglycan serological panel, and 20 CD-associated genetic variants were analyzed. Outcomes: complicated phenotype at diagnosis and early complicated course (hospitalizations/surgeries) within 2 years of diagnosis. Results: We included 260 patients (104, Indian (65.4%, male; age, 37.8); 156 Israeli (49.4%, male; 31.8, age)). Median lag time from symptoms onset to diagnosis was 10.5 (IQR 3–38) vs. 3 (IQR 1–8) months in Indian vs. Israeli patients (*p* < 0.001). Complicated phenotype at diagnosis was observed in 48% of Indian and 30% of Israeli patients (*p* = 0.003). Complicated phenotype was associated with higher anti-Saccharomyces cerevisiae antibody (ASCA) seropositivity rate among Israeli patients (*p* < 0.001), but not among Indian patients. Antiglycan serology did not correlate with the tested genetic variants. Early complicated course occurred in 28 (18%) Israeli and 13 (12.5%) Indian patients. The time from diagnosis to complication was comparable (log rank *p* = 0.152). Antiglycan serology did not correlate with a complicated early course in either cohort. Conclusions: There are significant differences in patients presenting with newly diagnosed CD in India and Israel, including phenotype and distinct biomarkers at diagnosis. These differences suggest different genetic and environmental disease modifiers.

## 1. Introduction

Crohn’s disease (CD) incidence and prevalence rates vary among different ethnic populations and geographic regions [1,2]. In recent years, there has been a global emergence of inflammatory bowel diseases (IBD), including CD, in previously less affected populations, including those of central and southern Europe, Southeast Asia, Africa, and Latin America [3]. Compared to other countries in Southeast Asia, there is a marked increase in the incidence and prevalence of CD in India [4,5,6,7,8]. While the prevalence of CD in India has not reached that of most Western countries [2], with an estimated population approaching 1.4 billion, the total number of cases in India is globally among the largest. The expected burden is concerning. Efforts to identify disease triggers and biomarkers of complications are thus specifically important.

Disease presentation, manifestations, and clinical course among Indian patients with CD may be different in comparison to what is currently known in the Western Hemisphere [9,10,11,12]. However, most of these data are derived from indirect comparisons and retrospective cohorts. More importantly, the biomarkers of disease course and risk stratification in the Indian CD population are scarce [12]. While some serologic responses to microbial antigens [13,14] and genetic variants [15] were found to correlate with complicated CD phenotype and course, their clinical utility for the Indian CD population remains largely unknown [16,17].

Facing the disparities between developing and developed countries, we initiated the Indo-Israeli IBD GastroEnterology paRtnership (TiiiGER), aiming to assess differences in disease presentation among patients with newly diagnosed CD in India and Israel, and to explore serotype–phenotype correlations.

## 2. Materials and Methods

### 2.1. Patient Selection

We initiated a prospective observational cohort in two large tertiary IBD referral centers: the Asian Institute of Gastroenterology (AIG) in Hyderabad, India, and the Tel Aviv Medical Center (TLVMC), Tel Aviv, Israel. We recruited consecutive adults (≥18 years old) that were newly diagnosed patients with CD between April 2013 and December 2018 at the AIG, and between September 2014 and March 2017 in TLVMC. Patients were recruited at the time of diagnosis and no later than 6 months from diagnosis. At both sites, CD diagnosis was confirmed on the basis of accepted clinical, laboratory, endoscopic, histologic, and imaging criteria. Disease phenotypes were classified according to the Montreal classification [18]. Patients were followed up for 24 months, and those who had fewer than 24 months of follow-up were excluded. Clinical data were collected prospectively during both standard and study follow-up visits.

### 2.2. Definitions and Outcomes

We addressed the following relevant complications to early CD: (a) *Complicated CD phenotype at diagnosis* was defined as stricturing and/or penetrating and/or perianal disease upon disease diagnosis; (b) *complicated early CD course* was defined clinically by the need for CD-related hospitalization and/or surgery (including intra-abdominal and perianal surgeries) within 2 years following diagnosis.

### 2.3. Serologic Analysis

Serum samples obtained at the time of recruitment were used for CD-associated antiglycan serologic analysis. We performed enzyme-linked immunosorbent assays (ELISA) on the blood samples using a commercially available kit (IBDx^®^, Glycominds Ltd., Lod, Israel) at both sites. Subjects were tested for the presence of antichitobioside carbohydrate (ACCA), antilaminaribioside carbohydrate (ALCA), antimannobioside carbohydrate (AMCA), and anti-*Saccharomyces cerevisiae* (ASCA) antibodies. For each antiglycan antibody, we documented both the absolute titer in IU/mL and serologic status (i.e., positive or negative). The cut-off values used for determining positive serologic response were AMCA > 100 IU/mL, ACCA > 90 IU/mL, ASCA > 50 IU/mL, and ALCA > 60 IU/mL based on the manufacturer’s instructions.

### 2.4. Genetic Analysis

Overall, 225/260 (86.5%) of the patients provided consent to conduct genetic analyses.

Upon recruitment, whole blood samples (3–6 mL) collected in standard EDTA collection tubes were provided. DNA was extracted from leukocytes using the FlexiGene DNA kit (Qiagen, Hilden, Germany). Quality and quantity were tested using a NanoDrop One spectrophotometer (Thermo Scientific, Waltham, MA, USA). To explore possible genetic biomarkers, we genotyped 20 CD-associated single-nucleotide polymorphisms (SNPs) that were reported in either Western (*n* = 13) or Indian (*n* = 7) cohorts in association with CD diagnosis [19,20,21] and course [22,23] (listed in Appendix A). These included polymorphisms in protein binding domain 2 (*NOD2*) [24] and the genes regulating the autophagic response to invading pathogens (autophagy pathway-related 16 like 1 (*ATG16L1*) and immunity-related GTPase 1 (*IRGM*)) [25]. Genotyping was performed using a TaqMan^®^ Gene Expression Assay (Applied Biosystems, Waltham, MA, USA), custom or predesigned, according to manufacturer instructions. Results were analyzed using StepOne software v2.3 (Applied Biosystems, Waltham, MA, USA).

### 2.5. Statistical Analysis

Categorical variables are expressed as frequency and percentage. Continuous variables were evaluated for normal distribution with a histogram, and are expressed as the mean and standard deviation (SD) or median and interquartile range (IQR). Follow-up time was observed with the reverse censoring method. Kaplan–Meier (KM) curves were created to evaluate disease complications during the follow-up period. All variables associated with disease complications were included in the multivariable Cox regression model.

To assess the association between each of the genetic variants with the four tested antiglycan antibodies, the chi-squared or Fisher exact test was applied. The false discovery method (FDR) was used to adjust the *p*-values for multiple comparisons.

All statistical tests were two-sided, and *p* < 0.05 was considered to be statistically significant. All statistical analyses were performed with SPSS software (IBM SPSS Statistics for Windows, version 25, IBM Corp., Armonk, NY, USA, 2017).

### 2.6. Ethical Statement

Signed informed consent was obtained from all participants at both sites. The study was approved by the institutional review boards of TLVMC (0467-10-TLV, September 2014) and AIG (ECR-346, April 2013).

## 3. Results

We included 260 patients with newly diagnosed CD (median age, 30.8; IQR: 23.1–41.7; males, 56.5%), of whom 104 were Indians, and 156 Israelis. All patients completed 2 years of follow-up. Table 1a depicts the baseline characteristics of the cohort, and Table 1b summarizes the baseline laboratory data.

Many distinctive baseline characteristics were observed between the cohorts. Indian, compared to Israeli, patients had male predominance (65.4% vs. 49.4%, *p* = 0.019) and were older (37.8 ± 12.8 vs. 31.8 ± 12.8, *p* < 0.001). Israeli, compared to Indian, patients were more likely to have a family history of IBD (22.4% vs. 3.8%, *p* < 0.001).

Disease location and phenotype at diagnosis were distinctive as well. Indian, compared to Israeli, patients had significantly more colonic (L2) and upper gastrointestinal disease location (L4) (35.6% vs. 19.2%, *p* = 0.004 and 13.5% vs. 5.1%, *p* = 0.018, respectively). Stricturing (B2) CD behavior was more common among Indian than that in Israeli patients (35.6% vs. 6.4%, *p* < 0.001), while perianal disease involvement was relatively rare among Indian patients (5.8% vs. 23.7% in Israelis, *p* < 0.001). Overall, patients in the Indian cohort were more frequently diagnosed with a complicated CD phenotype compared to patients in the Israeli cohort (48.1% vs 30.1%, *p* = 0.003).

The median lag time between symptoms onset and CD diagnosis was 3 (IQR 1–8) and 10.5 (IQR 3–38) months in the Israeli and the Indian cohorts, respectively (*p* < 0.001).

### 3.1. Distinctive Serologic Response to Antiglycans

We next explored whether serologic responses would be different in the Indian and Israeli cohorts at the time of CD diagnosis. To this end, we measured the expression of serologic markers in both groups. Serum samples obtained at diagnosis were available for 248/260 (95.4%) of the patients. Overall, 30/104 (28.8%) of the Indian patients and 53/144 (36.8%) of the Israeli patients had at least one positive antiglycan antibody (*p* = 0.18). Interestingly, there were significant differences not only in the proportion of patients with positive serologies, but also in the average titers of different antiglycan antibodies (Table 1b). Specifically, ASCA was positive in a significantly higher proportion of the Israeli cohort. Furthermore, average titers were approximately twice those in the Indian cohort. Additionally, the proportion of Indian patients positive for ALCA and ACCA was significantly higher than that in the Israeli cohort, whereas AMCA positivity was similar between the cohorts.

As serologic profiles may represent different disease locations [13,26], we next correlated antiglycan expression with disease location. In the Israeli cohort, ASCA positivity was more associated with an ileocolonic than with a colonic location (L3 vs. L2, 46.8% vs. 10.6%, *p* = 0.013), but did not differ between ileal and colonic (L1 vs. L2, *p* = 0.17), and ileal and ileocolonic (L1 vs. L3, *p* = 0.12). Similarly, 80% of ACCA-positive Israeli patients had ileocolonic CD vs. 20% and 0% of those with ileal and colonic locations (*p* = 0.046). In contrast, in the Indian cohort, none of the antibodies’ seropositive status correlated with CD location.

### 3.2. Assessing Factors Associated with Complicated Phenotype at Diagnosis

Comparing the age at time of diagnosis, sex, smoking status, and family history of IBD between patients with complicated compared to uncomplicated phenotype at presentation revealed no baseline differences (Appendix A). Since longer diagnostic lag time may result in progressive bowel damage, we next analyzed the relationship between diagnostic lag time and stricturing/penetrating phenotypes (B2/B3) at diagnosis. There were no differences between the diagnostic lag time of patients diagnosed with B1 compared to B2/B3 phenotypes within each cohort (*p* = 0.43, Israeli cohort; *p* = 0.99, Indian cohort).

To explore biomarkers for a complicated CD phenotype, we sought associations between a complicated-CD phenotype at diagnosis and antiglycan antibody expression (Table 2). Interestingly, of the four tested antiglycan antibodies, ASCA seropositivity at diagnosis was more associated with a complicated than with noncomplicated CD phenotype among Israeli patients (52.2% vs. 23.5%, *p* = 0.001). In contrast, in the Indian cohort, none of the antibodies’ seropositive status correlated with a complicated CD-phenotype (Table 2).

### 3.3. Assessing Factors Associated with Early Complicated Course

Notably, 21.1% of Indian patients have received empirical antituberculosis treatment (ATT). Since empirical ATT therapy might delay the initiation of an effective IBD-specific therapy and further contribute to bowel damage we then correlated ATT use with disease phenotype at diagnosis. No significant difference was noted between Indian patients diagnosed with a B2/B3 phenotype compared to those with B1 phenotype (*p* = 0.2).

CD treatment in the Israeli and the Indian cohorts during the first 2 years is shown in Table 1b. Induction with corticosteroids and maintenance with immunomodulators were similarly used in the two cohorts (47.4% and 55.8%, *p* = 0.187 and 44.2% and 51.9%, *p* = 0.223 for Israeli and Indian patients, respectively). In contrast, significant differences in using 5-ASA and biological therapy were observed. Specifically, 5-ASA agents were prescribed to 95.2% of the Indian compared to 38.5% of the Israeli cohort (*p* < 0.001). Biological therapy was administered to 41.7% of the Israeli cohort, but only to 5.8% of the Indian cohort (*p* < 0.001). In both cohorts, the first-line biologic therapy was antitumor necrosis factor (TNF)-alpha. In the Indian cohort, immunomodulators were used as monotherapy in the vast majority of cases (92.5%), but in the Israeli cohort, only 14.5% (*p* < 0.001) of patients were treated with monotherapy immunomodulators, while the remaining received immunomodulators in combination with anti-TNF-alpha agents.

During the 2-year follow-up, 28 (17.9%) Israeli and 13 (12.5%) Indian patients required either CD-related hospitalization or surgery and thus were considered to have a complicated early CD course. Time to complication in both cohorts was comparable (Kaplan–Meier analysis log rank *p* = 0.152, Figure 1A).

Then, we explored demographic, clinical, serologic, and genetic factors in association with early complicated CD course in each cohort. The baseline demographics of age at diagnosis, family history of IBD, body mass index, smoking status, and baseline hemoglobin level were not associated with an early complicated CD course within either cohort (data not shown). Of all variables, only NOD2 variant rs2066847 was associated with an early complicated course (*p* < 0.001) in Indian patients (Figure 1B). None of the genetic variants was associated with a complicated disease course in Israeli patients. While the clinical presentation did not correlate with early disease complications within the Indian cohort, in multivariable analysis, stricturing/penetrating phenotype (HR 2.4, 95% CI:1–5.6, *p* = 0.041) and perianal involvement (HR 2.471, 95% CI:1–5.7, *p* = 0.037) at diagnosis correlated with early complicated CD course in the Israeli cohort (Figure 1C,D).

### 3.4. No Genotype–Serotype Interactions Were Identified

Previous data from our group demonstrated that genetic variants may have disease-specific serologic effects [27]. To further investigate whether the differential serologic response is related to specific genetic predisposition, we employed genetic analysis and genetic–serologic correlation. Out of the 20 analyzed SNPs, 7 were differentially expressed between the cohorts. Specifically, 17.2% of the Israelis and 1.9% of the Indians were carriers of the NOD2_rs2066845 variant (*p* < 0.001). In contrast, 60.2% of the Indians carried the TNF-α_rs1799964 variant compared to 32.7% of the Israelis (*p* < 0.001) (Table 3). For genotype–serotype interaction assessment, we correlated the carriage status of each genetic variant (defined as either heterozygous or homozygote state) with the seropositivity rate status of each antiglycan antibody. Associations were then corrected to multiple comparisons. No genetic-serologic correlation was noticed (data not shown).

## 4. Discussion

In this study, we aimed to decipher similarities and differences between Indian and Israeli patients with newly diagnosed CD. We thus compared two cohorts of Indian and Israeli patients with newly diagnosed CD that were followed prospectively through the first two years after diagnosis. We demonstrated strikingly different clinical presentation of CD in diverse ethnic populations, specifically disease location and phenotype, with a more frequently complicated phenotype at diagnosis in the Indian cohort compared to the Israeli cohort. Serologic differences were noticed; specifically, ASCA was more prevalent in the Israeli cohort, and ACCA and ALCA were more prevalent in the Indian cohort. ASCA correlated with complicated phenotype at diagnosis only in the Israeli cohort, and no correlations were identified in the Indian cohort. Lastly, distinctive clinical predictors and biomarkers for early complicated CD course were found in Indian patients (specifically, *NOD2* variant rs2066847), and in Israeli patients (specifically, complicated phenotype at diagnosis).

A higher rate of a complicated phenotype at diagnosis was noticed in Indian patients compared to that in the Israeli cohort, driven predominantly by the B2 phenotype. This proportion was significantly higher than that of the Israeli cohort and other Western cohorts [28,29]. While ethnicity may influence CD phenotype at diagnosis [30,31], the relatively new emergence of IBD in Asia may lead to multiple barriers and a lack of awareness, leading to diagnostic delay. Within each cohort, we did not detect an association between diagnostic delay and complicated phenotype at diagnosis, but symptom onset might poorly correlate with inflammatory disease activity in CD [32] and thus with biological disease onset. In general, the Indian cohort demonstrated a significantly longer lag time between symptom onset to diagnosis and older age at time of diagnosis, both of which may support a possible role of diagnostic delay in the complicated phenotype. Additionally, the empirical use of antitubercular therapy, which was used in more than 40% of the Indians and none of the Israelis, was found to contribute to the stricturing phenotype at CD diagnosis among Indian patients [12].

We analyzed a set of antiglycan antibodies associated with CD as potential proxies assumed to reflect genetic–environmental interaction [33,34,35]. Some of these antibodies were specifically associated with CD diagnosis and complications in several studies [13,14,36]. Antiglycan serologic responses in Indian patients with CD [17,18,19,20,21,22,23,24,25,26,27,28,29,30,31,32,33,34,35,36,37] and non-Indian Asian cohorts [38,39] were scarcely reported. Interestingly, despite a comparable overall rate of seropositivity, differences between the cohorts were noticed. Notably, ALCA and ACCA positivity characterized more the Indian than the Israeli cohort, while ASCA was expressed in one-third of the Israeli cohort compared to only 13.5% in the Indian cohort. AMCA rate was similar. Our findings contrast with earlier studies examining the expression of ASCA in Asian CD cohorts from Japan [39], Hong Kong [40], and Korea [41], showing similar rates of ASCA seropositivity to those reported in Western cohorts. Variability in the presence of antiglycans may result from differences in bowel microbial composition, dietary exposure, the degree of urbanization, or differential genetic predisposition. Serologic characteristics may indicate a distinctive phenotype not only in the Indian compared to the Israeli cohort, but also to within-Asia differences, specifically between the Hindu populations included in the AIG cohort, and those with Japanese and Chinese ethnic backgrounds. To explore whether genetic factors contributed to the latter, we investigated several genetic variants related to the innate immune system in genes that are involved in pattern recognition and autophagy (NOD2, IRGM, and ATG16l1) and associated with CD. Furthermore, some of these genetic variants were previously found to influence serologic expression in patients with CD [27,42]. No correlation remained significant after controlling for multiple comparisons. In order to decipher whether the noticed differences are meaningful, larger cohorts may be needed.

We also explored the role of serology as a biomarker for complicated phenotypes at diagnosis. Indeed, ASCA positivity correlated well with a complicated CD phenotype in Israeli patients, corroborating previous observations from Western cohorts [13,14]. Conversely, in the Indian cohort, no association between any of the tested serologic markers with complicated phenotypes was observed. These interesting differences support discordant patient–environment interactions in the two evaluated populations. Furthermore, they suggest that environmental modifiers may contribute to variable complications and phenotypes. Further research into those modifiers may contribute not only to understanding of disease pathophysiology, but also to its treatment and control.

The prediction of CD course is challenging. We demonstrated distinctive predictors for early complicated CD in the Israeli and the Indian cohorts. Specifically, the genetic variant in NOD2, rs2066847, was associated with a complicated course among the Indian group. Interestingly, no such association was demonstrated in the Israeli cohort. Genetic CD modification was previously reported in Western cohorts [43] specifically regarding NOD rs2066847 variants [44]. The observation in Indian populations is less recognized. Our findings highlight the central role that NOD rs2066847 may have as a disease modifier. Interestingly, this variant had been identified in association with complicated disease course of CD in Western cohorts [43,44]. To the best of our knowledge, no previous reports addressed whether NOD2-positive CD in Indian patients is a risk factor for a more aggressive CD course. In the Israeli cohort, none of the genetic variants was associated with complicated course, possibly due to the relatively small sample and the limited number of genetic variants that were tested.

Our study has several strengths. We could directly compare two cohorts of newly diagnosed patients in India and the West, and to show the major differences in presentation and early disease course. Comparison of prospective cohorts assessing early CD course was scarcely reported. Ng et al. compared the epidemiology and phenotype of IBD in 8 Asian countries and Australia diagnosed between 2011 and 2012 [45]. This pivotal study highlighted the potential severity of IBD in Asians countries; however, the assessment of disease-specific biomarkers was not included. Furthermore, data from centers in India were not included either. Currently, most of the data on similarities and differences among CD genetics, risk factors, and course are based on indirect comparisons [5,10,46]. To the best of our knowledge, our study is the first to explore the use of antiglycan antibodies in Indian patients with CD in association with a complicated phenotype and course. Furthermore, similarly performing exploratory biomarker analyses in both cohorts contributes to the validity of outcomes.

We acknowledge several limitations: while recruiting over 100 patients in each inception cohort, the absolute number of patients may have limited our ability to assess genotype–serotype interactions. Notably, the size of the recruited inception cohorts still enabled indicating differences in presentation and phenotype at diagnosis. Follow-up lasted 2 years after diagnosis; therefore, predictors for longer-term outcomes could not be assessed. The exploratory genetic analysis included 20 selected SNPs, potentially overlooking serotype–genotype interactions in other loci, and further being limited by the number of tested individuals. The definition of a complicated CD course that we used, which included hospitalization and surgery, was as previously performed [47,48,49]. However, complications such as steroidal dependency use were not included. Lastly, the inherent differences in the medical systems and environmental exposure between the cohorts (such as dietary exposure and environmental hygiene) may limit direct comparison. To overcome this potential limitation, we separately analyzed for associations between serologic and clinical features and serotype–genotype interactions within each cohort.

While the absolute number of patients living with IBD in India is expected to reach that of the United States [50], biomarkers and predictors for complicated CD course in the Indian population are largely unknown. On the basis of two inception cohorts in Israel and India, and a combination of clinical, serologic, and genetic analyses, we demonstrated a high proportion of complicated CD at diagnosis in the Indian population, a differential antiglycan serologic response, and a lack of correlation with a complicated phenotype among Indians vs. the correlation with a complicated phenotype among Israelis. Lastly, our findings suggest that predictors on an early complicated course are different in different populations. Altogether, our results highlight unique characteristics of early CD course in genetically and environmentally diverse populations.

## Figures and Tables

**Figure 1 jcm-11-06899-f001:**
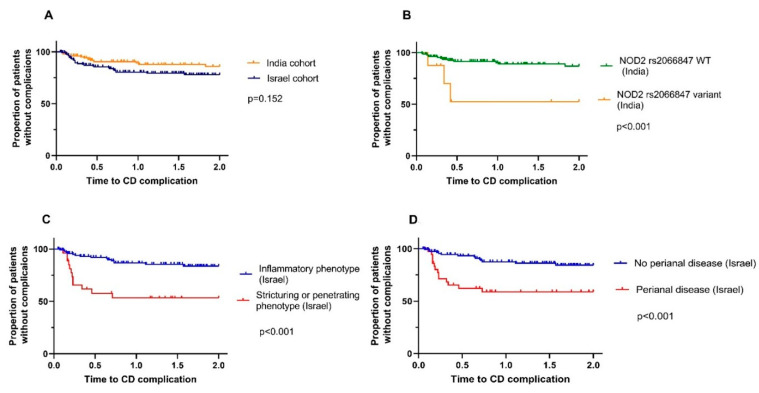
Kaplan–Meier plots showing the proportion of patients with newly diagnosed Crohn’s without complication, stratified by the following: (**A**) Indian and Israeli cohorts; (**B**) NOD2 rs2066847 variant in Indian cohort; (**C**) inflammatory vs stricturing and/or penetrating phenotype in Israeli cohort; (**D**) perianal disease in Israeli cohort.

**Table 1 jcm-11-06899-t001:** (a) Demographic and clinical characteristics of Indian and Israeli cohorts (*n* = 260). (b) Biochemical characteristics and antiglycan serologic responses magnitude and seropositivity rates at the presentation of the Indian and Israeli cohorts (*n* = 260).

**(a)**	**Indian Cohort (*n* = 104)**	**Israeli Cohort (*n* = 156)**	** *p* ** **-Value**
Sex, male, *n* (%)	68 (65.4)	77 (49.4)	0.019
Age, years ±SD	37.8 ± 12.8	31.8 ± 12.9	<0.001
BMI, kg/m^2^ ± SD	19.9 ± 4.3	23.4 ± 4.5	<0.001
Active smokers, *n* (%)	9 (8.7)	34 (21.8)	<0.001
Family history of Crohn’s disease, *n* (%)	4 (3.8)	35 (22.4)	<0.001
Extraintestinal manifestations, *n* (%)	22 (21.2)	53 (39.6)	0.003
Treatment			
Corticosteroids, *n* (%)	58 (55.8)	74 (47.4)	0.187
5-ASA, *n* (%)	99 (95.2)	60 (38.5)	<0.001
IMM, *n* (%)	54 (51.9)	69 (44.2)	0.223
Biologic therapy, *n* (%)	6 (5.8)	65 (41.7)	<0.001
Disease location			
Ileal, *n* (%) ^a^	45 (43.3)	70 (44.9)	0.004
Colonic, *n* (%) ^b^	37 (35.6)	30 (19.2)	
Ileocolonic, *n* (%) ^c^	22 (21.2)	55 (35.3)	
Upper GI involvement, *n* (%)	14 (13.5)	8 (5.1)	0.018
Perianal, *n* (%)	6 (5.8)	37 (23.7)	<0.001
Disease phenotype			
Inflammatory, *n* (%) ^d^	56 (53.8)	129 (82.7)	<0.001
Stricturing, *n* (%) ^e^	37 (35.6)	10 (6.4)	
Penetrating, *n* (%) ^f^	11 (10.6)	17 (10.9)	
Complicated phenotype at diagnosis			
Stricturing and/or penetrating and/or perianal, *n* (%)	50 (48.1)	47 (30.1)	0.003
Lag time from symptoms to diagnosis, months, median (IQR)	10.5 (3–38)	3 (1–8)	*p* < 0.001
**(b)**	**Indian Cohort (*n* = 104)**	**Israeli Cohort (*n* = 156 *)**	** *p* ** **-Value**
Hemoglobin, g/dL, median (IQR)	11.3 (9.9–12.8)	13.1 (12.1–14.3)	<0.001
White blood cells, K/uL, median (IQR)	7.9 (6.0–9.4)	8.2 (6.8–10.2)	0.019
C-reactive protein, mg/L, median (IQR)	9.3 (5.4–15.5)	9.9 (2.5–21.6)	0.447
Vitamin B12, pg/mL, median (IQR)	298 (213–479)	393 (309–522)	<0.001
ASCA, IU/mL, mean ± SD	24.2 ± 21.3	43.0 ± 39.0 *	<0.001
ALCA, IU mean ± SD	26.6 ± 23.6	18.2 ± 14.5 *	0.005
ACCA, IU/mL, mean ± SD	41.4 ± 36.9	31.5 ± 24.9 *	0.396
AMCA, IU mean ± SD	43.1 ± 42.3	57.2 ± 61.1 *	<0.001
ASCA-positive, *n* (%)	14 (13.5)	47 (32.6) *	0.001
ALCA-positive, *n* (%)	13 (12.5)	3 (2.1) *	0.001
ACCA-positive, *n* (%)	18 (17.3)	5 (3.5) *	<0.001
AMCA-positive, *n* (%)	13 (12.5)	15 (10.3) *	0.685
Any positive serology, *n* (%)	30 (28.8)	53 (36.8) *	0.180

SD, standard deviation; *n*, number; SD, standard deviation; 5-ASA, 5-aminosalycilate; IMM, immune modulators; GI, gastrointestinal. *p*-values for comparison between the two groups: a-*p* = 0.899, b-*p* = 0.004, c-*p* = 0.018, d-*p* < 0.001, e-*p* = <0.001, f-*p* = 0.934. * Serologic data were applicable for 144 of the 156 patients of the Israeli cohort. *n*, number; IQR, interquartile range; ASCA, anti-Saccharomyces cerevisiae antibody; ALCA, antilaminaribioside carbohydrate antibody; ACCA, antichitobioside carbohydrate antibody; AMCA, antimannobioside carbohydrate antibody.

**Table 2 jcm-11-06899-t002:** Comparison of positive antiglycan serologic responses between patients with newly diagnosed complicated and uncomplicated CD phenotypes in Indian and Israeli cohorts.

	Indian Cohort (*n* = 104)			Israeli Cohort (*n* = 144 *)		
	Complicated Phenotype ^&^ (*n* = 50)	Uncomplicated Phenotype (*n* = 54)	*p* Value	Complicated Phenotype ^&^ (*n* = 46)	Uncomplicated Phenotype (*n* = 98)	*p* Value
Any positive serology, *n* (%)	13 (26.0)	17 (31.5)	0.665	24 (52.2)	29 (29.6)	0.01
ASCA-positive serology, *n* (%)	8 (16)	6 (11.1)	0.570	24 (52.2)	23 (23.5)	0.001
ALCA-positive serology, *n* (%)	5 (10)	8 (14.8)	0.559	1 (2.2)	2 (2)	0.999
ACCA-positive serology, *n* (%)	7 (14)	11 (20.4)	0.445	3 (6.5)	2 (2)	0.327
AMCA-positive serology, *n* (%)	3 (6)	10 (18.5)	0.075	8 (17.4)	7 (7.1)	0.08

*n*, number; ASCA, anti-Saccharomyces cerevisiae antibody; ALCA, antilaminaribioside carbohydrate antibody; ACCA, antichitobioside carbohydrate antibody; AMCA, antimannobioside carbohydrate antibody; ^&^ Complicated phenotype included stricturing and/or penetrating disease and/or perianal involvement at diagnosis. * Serologic analysis included 144 of 156 patients in the Israeli cohort.

**Table 3 jcm-11-06899-t003:** Susceptible genes variation in Indian and Israeli cohorts.

Genetic Variant	Israeli Cohort (*n* = 122)	Indian Cohort (*n* = 103)	*p* Value
ATG16L1_rs2241880, *n* (%)			
Wild type	16 (13.1)	25 (24.3)	0.061
Heterozygous	59 (48.4)	49 (47.6)	
Homozygous	47 (38.5)	29 (28.2)	
CARD8_rs2043211, *n* (%)			
Wild type	65 (53.3)	42 (40.8)	0.17
Heterozygous	46 (37.7)	49 (47.6)	
Homozygous	11 (9.0)	12 (11.7)	
CARD9_rs10781499, *n* (%)			
Wild type	48 (39.3)	39 (37.9)	0.9
Heterozygous	58 (47.5)	52 (50.5)	
Homozygous	16 (13.1)	12 (11.7)	
NOX3_rs6557421, *n* (%)			
Wild type	53 (43.4)	59 (57.3)	0.11
Heterozygous	61 (50.0)	40 (38.8)	
Homozygous	8 (6.6)	4 (3.9)	
NOD2_rs2066844, *n* (%)			
Wild type	117 (95.9)	103 (100)	0.064
Heterozygous	5 (4.1)	0 (0)	
Homozygous	0 (0)	0 (0)	
NOD2_rs2066845, *n* (%)			
Wild type	101 (82.8)	101 (98.1)	<0.001
Heterozygous	21 (17.2)	2 (1.9)	
Homozygous	0 (0)	0 (0)	
NOD2_rs2066847, *n* (%)			
Wild type	113 (92.6)	95 (92.2)	0.052
Heterozygous	9 (7.4)	4 (3.9)	
Homozygous	0 (0)	4 (3.9)	
IRGM_rs11741861, *n* (%)			
Wild type	70 (57.4)	62 (60.2)	0.926
Heterozygous	45 (36.9)	35 (34.0)	
Homozygous	7 (5.7)	6 (5.8)	
IRGM_rs4958847, *n* (%)			
Wild type	62 (50.8)	46 (44.7)	0.2
Heterozygous	51 (41.8)	42 (40.8)	
Homozygous	9 (7.4)	15 (14.6)	
MHC_ rs9279411, *n* (%)			
Wild type	117 (95.9)	103 (100)	0.064
Heterozygous	5 (4.1)	0 (0)	
Homozygous	0 (0)	0 (0)	
XACT _rs5929166, *n* (%)			
Wild type	121 (99.2)	102 (99)	0.999
Heterozygous	1 (0.8)	1 (1.0)	
Homozygous	0 (0)	0 (0)	
IGFBP_rs75764599, *n* (%)			
Wild type	116 (95.1)	103 (100)	0.032
Heterozygous	6 (4.9)	0 (0)	
Homozygous	0 (0)	0 (0)	
FOXO3_rs147856773, *n* (%)			
Wild type	92 (75.4)	81 (78.6)	0.815
Heterozygous	29 (23.8)	21 (20.4)	
Homozygous	1 (0.8)	1 (1.0)	
ATG16L1_rs4663402, *n* (%)			
Wild type	116 (95.1)	88 (85.4)	0.028
Heterozygous	5 (4.1)	13 (12.6)	
Homozygous	1 (0.8)	2 (1.9)	
ATG16L1_rs4663421, *n* (%)			
Wild type	116 (95.1)	88 (85.4)	0.028
Heterozygous	5 (4.1)	13 (12.6)	
Homozygous	1 (0.8)	2 (1.9)	
IRGM_rs1000113, *n* (%)			
Wild type	70 (57.4)	63 (61.2)	0.838
Heterozygous	45 (36.9)	34 (33)	
Homozygous	7 (5.7)	6 (5.8)	
IRGM_rs180802994, *n* (%)			
Wild type	118 (96.7)	95 (92.2)	0.150
Heterozygous	4 (3.2)	8 (7.8)	
Homozygous	0 (0)	0 (0)	
TNF-α_rs1799964, *n* (%)			
Wild type	82 (67.2)	41 (39.8)	<0.001
Heterozygous	33 (27)	51 (49.5)	
Homozygous	7 (5.7)	11 (10.7)	
TNF-α_rs1799724, *n* (%)			
Wild type	79 (64.8)	81 (78.6)	0.012
Heterozygous	37 (30.3)	22 (21.4)	
Homozygous	6 (4.9)	0 (0)	
JAK2_rs1887428, *n* (%)			
Wild type	49 (40.2)	28 (27.2)	0.050
Heterozygous	55 (45.1)	49 (47.6)	
Homozygous	18 (14.8)	26 (25.2)	

## Data Availability

The data presented in this study are available on request from the corresponding author. The data are not publicly available.

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
