# Peer review of "Newly Diagnosed Crohn’s Disease Patients in India and Israel Display Distinct Presentations and Serological Markers: Insights from Prospective Cohorts"

_jcm, 2022, doi:10.3390/jcm11236899_

Round 1

Reviewer 1 Report

The manuscript entitled “Newly Diagnosed Crohn's Disease Patients in India and Israel 2 Display Distinct Presentations and Serological Markers: In- 3 sights from East-West Cohorts” by Goren et al. compared Crohn’s disease (CD) patients cohort from two different regions and highlighted the similarities and differences in various features such as sero-positivity, genotype and complexity related to the disease pathology.  

This study is carefully designed, nicely presented and provides the valuable information about the CD from two different geographic regions.

Minor comment:

Supplementary table 3: It would be interesting if authors split this table and include key data sets in the main manuscript for the ease of the readers.

Author Response

We thank the reviewer for the positive feedback. As requested, we moved the table presenting the genetic variants into the main manuscript. In addition, the key genetic findings are presented in the results section.  

Reviewer 2 Report

In this paper “Newly Diagnosed Crohn’s Disease Patients in India and Israel Display Distinct Presentations and Serological Markers: Insights from East-West Cohorts” the Authors discuss the differences in clinical presentation, serology, genetic expression and clinical course between patients with newly diagnosed Crohn’s disease (CD) from two different geographic areas, India and Israel, by conducting a prospective study with a follow-up time of two years.

The design of the study is valid and consistent with its purpose, however there are some critical issues needing to be addressed:

-       In the title and in several parts of the paper (e.g. page 2, line 71; page 11, line 351) the Authors tend to extend their findings to the dichotomy between East and West of the World, but this appears to be incorrect since Israel is better defined as a Middle East country and since its peculiar IBD epidemiology, which is very different in Jewish and Arab patients.

-       There is no clear definition of what is considered by the Authors a “newly” diagnosed Crohn’s disease; this should be clarified in the section “Material and Methods”

Other minor comments:

-       Page 9, figure n° 1: plots report on the y-axis “Probability of Survival" but there is no reference to this topic in the paper

-       Page 10, line 271: p value regarding the different expression of the variant “TNF-α_rs1799964” between the two cohorts is missing

-       Page 10, line 298: closing parenthesis is missing

-       Some of the references are reported twice: reference n° 5 is identical to n° 52, n° 10 to n° 51

The results of this paper could be less appealing to physicians of other states, whereas they will certainly be interesting to the clinical practitioners working in India, as they show some peculiarities of Indian population with CD, which are even more important in the era of “personalized medicine”, although at the moment genetic and serological analysis have limited impact on clinical practice.

Author Response

The design of the study is valid and consistent with its purpose.

Response - We thank the reviewer for the positive feedback.

In the title and in several parts of the paper (e.g. page 2, line 71; page 11, line 351) the Authors tend to extend their findings to the dichotomy between East and West of the World, but this appears to be incorrect since Israel is better defined as a Middle East country and since its peculiar IBD epidemiology, which is very different in Jewish and Arab patients.

Responsewe agree with the reviewer that Israel is geographically located in the Middle East, however, it shares multiple IBD-related characteristics with the Western hemisphere (i.e. high prevalence, urbanization, healthcare system, etc). To comply with the reviewer's request we omitted the "East-West cohorts" from the title and modified the aim section to comparison between "developing and developed countries".  

There is no clear definition of what is considered by the Authors a “newly” diagnosed Crohn’s disease; this should be clarified in the section “Material and Methods”

Responsewe thank the reviewer for this comment. Consecutive patients were recruited at the time of diagnosis and up to 6 months from diagnosis at both sites. These data are now specified in the methods section.  

Page 9, figure n° 1: plots report on the y-axis “Probability of Survival" but there is no reference to this topic in the paper

Response – the y-axis was modified to "Proportion of patients without complication" and is referred to in the manuscript (page 8, line 249).

Page 10, line 271: p-value regarding the different expression of the variant “TNF-α_rs1799964” between the two cohorts is missing

Response – the missing P value (p<0.001) was added

Page 10, line 298: closing parenthesis is missing.

Response – the sentences was corrected

Some of the references are reported twice: reference n° 5 is identical to n° 52, n° 10 to n° 51

Response – we thank the reviewer. The references were corrected accordingly

The results of this paper could be less appealing to physicians of other states, whereas they will certainly be interesting to the clinical practitioners working in India, as they show some peculiarities of Indian population with CD, which are even more important in the era of “personalized medicine”, although at the moment genetic and serological analysis have limited impact on clinical practice.

Response - We thank the reviewer for the interest in our paper, and for acknowledging its contribution to the field. While the peculiarities of the Indian CD population might be less appealing to the clinical readership outside India, it highlights the global burden of IBD in developing nations. Studying the different interactions between genetics and the environment (serology) can shed light of differences in disease triggers and modifiers. More importantly, developing countries where CD incidence has not yet reached its peak, may implement abatement measures to reduce potential exposures, and prioritizing preventative measures.